# Dityrosine Crosslinking of Collagen and Amyloid-β Peptides Is Formed by Vitamin B_12_ Deficiency-Generated Oxidative Stress in *Caenorhabditis elegans*

**DOI:** 10.3390/ijms222312959

**Published:** 2021-11-30

**Authors:** Kyohei Koseki, Aoi Yamamoto, Keisuke Tanimoto, Naho Okamoto, Fei Teng, Tomohiro Bito, Yukinori Yabuta, Tsuyoshi Kawano, Fumio Watanabe

**Affiliations:** 1The United Graduate School of Agricultural Sciences, Tottori University, Tottori 680-8553, Japan; kyouhei.ganbalu@gmail.com (K.K.); mizonbon26@gmail.com (N.O.); yabuta@tottori-u.ac.jp (Y.Y.); kawano@tottori-u.ac.jp (T.K.); watanabe@tottori-u.ac.jp (F.W.); 2Department of Agricultural Science, Graduate School of Sustainability Science, Tottori University, Tottori 680-8553, Japan; m21j7037b@edu.tottori-u.ac.jp; 3Department of Agricultural, Life and Environmental Sciences, Faculty of Agriculture, Tottori University, Tottori 680-8553, Japan; b18a5103b@edu.tottori-u.ac.jp; 4Department of Food Quality and Safety, College of Food Science, Northeast Agricultural University, Harbin 150030, China; tengfei@neau.edu.cn

**Keywords:** Alzheimer’s disease, ascorbic acid, *Caenorhabditis elegans*, collagen, dityrosine crosslinking, oxidative stress, vitamin B_12_ deficiency

## Abstract

(1) Background: Vitamin B_12_ deficiency in *Caenorhabditis elegans* results in severe oxidative stress and induces morphological abnormality in mutants due to disordered cuticle collagen biosynthesis. We clarified the underlying mechanism leading to such mutant worms due to vitamin B_12_ deficiency. (2) Results: The deficient worms exhibited decreased collagen levels of up to approximately 59% compared with the control. Although vitamin B_12_ deficiency did not affect the mRNA expression of prolyl 4-hydroxylase, which catalyzes the formation of 4-hydroxyproline involved in intercellular collagen biosynthesis, the level of ascorbic acid, a prolyl 4-hydroxylase coenzyme, was markedly decreased. Dityrosine crosslinking is involved in the extracellular maturation of worm collagen. The dityrosine level of collagen significantly increased in the deficient worms compared with the control. However, vitamin B_12_ deficiency hardly affected the mRNA expression levels of *bli-3* and *mlt-7*, which are encoding crosslinking-related enzymes, suggesting that deficiency-induced oxidative stress leads to dityrosine crosslinking. Moreover, using GMC101 mutant worms that express the full-length human amyloid β, we found that vitamin B_12_ deficiency did not affect the gene and protein expressions of amyloid β but increased the formation of dityrosine crosslinking in the amyloid β protein. (3) Conclusions: Vitamin B_12_-deficient wild-type worms showed motility dysfunction due to decreased collagen levels and the formation of highly tyrosine-crosslinked collagen, potentially reducing their flexibility. In GMC101 mutant worms, vitamin B_12_ deficiency-induced oxidative stress triggers dityrosine-crosslinked amyloid β formation, which might promote its stabilization and toxic oligomerization.

## 1. Introduction

Vitamin B_12_ (B_12_) functions as the two coenzymes 5′-deoxyadenosylcobalamin and methylcobalamin of methylmalonyl-CoA mutase (EC 5.4.99.2) [1] and methionine synthase (MS; EC 2.1.1.13) [2], respectively, in mammals. Individuals deficient in B_12_ reportedly showed a significant increase of intracellular homocysteine (Hcy), a potent prooxidant [3], due to the reduced activity of MS, catalyzing methionine synthesis from Hcy and N5′-methyltetrahydrofolate [4]. Severe B_12_ deficiency leads to various symptoms, such as megaloblastic anemia, infertility, and neuropathy [3]. However, the underlying disease mechanisms are not fully understood [5,6].

As its molecular and cellular processes are similar to those of humans, *Caenorhabditis elegans* has been widely used as a model organism for genetic and biochemical studies. Our preceding study of B_12_ deficiency using *C. elegans* showed the occurrence of B_12_-deficient worms with specific morphological abnormalities, like the short and plump “dumpy” mutant phenotype induced by the disordered biosynthesis of cuticular collagen, the main extracellular matrix component of the worm cuticle [7]. Mammalian collagen biosynthesis is well known to involve various posttranslational modifications [8], such as proline hydroxylation of collagen polypeptide chains in the rough endoplasmic reticulum and lysine oxidation of collagen triple helices in the extracellular space, catalyzed by prolyl 4-hydroxylase (EC 1.14.11.2) [9] and lysyl oxidase (EC 1.14.11.4) [10], respectively. Lysyl oxidase-derived linkages are predominantly formed in mammals during the extracellular maturation of the collagen molecule. However, lysyl crosslinking is absent from the *C. elegans* cuticle collagen [11]: it is replaced by dityrosine crosslinking [12]. No evidence has indicated whether B_12_ deficiency could affect the intracellular biosynthesis and subsequent extracellular maturation of the worm cuticle collagen.

In this study, we demonstrated that B_12_ deficiency results in significantly decreased collagen levels due to decreased ascorbic acid, a prolyl 4-hydroxylase coenzyme, and increased dityrosine crosslinking formation, leading to motility dysfunction. Furthermore, to the best of our knowledge, this study is the first to report that the dityrosine crosslinking of collagen was induced by oxidative stress generated by B_12_ deficiency. The reactive oxygen species that induce dityrosine crosslinking are reportedly formed in the amyloid-β (Aβ) oligomers involved in Alzheimer’s disease (AD) pathogenesis [13]. Therefore, our finding could be applied to evaluate whether B_12_ deficiency could promote AD development. Using GMC101 mutant worms producing Aβ peptides in their muscle cells, we also discussed how B_12_ deficiency could affect the dityrosine crosslinking level of the Aβ peptide in inducing oligomerization and toxicity.

## 2. Results

### 2.1. Effect of B_12_ Deficiency on Collagen Biosynthesis in C. elegans

Figure 1A shows how B_12_ deficiency affected worm body collagen levels, calculated from the hydroxyproline content determined by the amino acid analysis of hydrolyzed worm body proteins. The worm collagen level significantly decreased during B_12_ deficiency. The decreased collagen level (approximately 8.1 ± 0.7 mg/g wet weight) was approximately 59% of that of control worms. B_12_ supplementation of B_12_-deficient worms completely recovered the reduced collagen level to that of the control (approximately 14.8 ± 0.4 mg/g wet weight).

To clarify the underlying mechanism of significantly reduced collagen levels during B_12_ deficiency, the prolyl 4-hydroxylase α (PHY-1 and PHY-2) and β (PDI) subunits’ mRNA expressions were determined. As shown in Figure 1B, B_12_ deficiency did not affect the prolyl 4-hydroxylase α (*dpy-18* and *phy-2*) and β (*pdi-2*) subunits’ mRNA expressions.

Prolyl 4-hydroxylase requires *L*-ascorbic acid (AsA) as a coenzyme. Therefore, the AsA level was determined in the homogenates of the control and B_12_-deficient worms (Figure 1C). B_12_ deficiency significantly reduced the AsA level (approximately 22.3 ± 5.2 µg/g wet weight), reaching approximately 52% of that of the control worms. B_12_ supplementation of B_12_-deficient worms showed that the reduced AsA levels were completely recovered to that of the control (approximately 45.3 ± 2.6 µg/g wet weight). These results indicated that the significantly decreased collagen biosynthesis in B_12_-deficient worms was mainly due to the reduced level of AsA as a coenzyme of prolyl 4-hydroxylase, as the mRNA expression levels of the enzyme subunits were completely unaffected by the B_12_ deficiency. 

### 2.2. Effect of B_12_ Deficiency on the Dityrosine Crosslinking Level of Worm Collagen

To clarify whether B_12_ deficiency affects dityrosine crosslinking in the extracellular maturation of cuticular collage, dityrosine levels were assayed in B_12_-deficient worms. Dityrosine levels (per mg collagen) significantly increased in the B_12_-deficient worms compared with control worms (Figure 2A). These results indicate that, although B_12_ deficiency significantly reduces collagen levels, the dityrosine crosslinking level of collagen was high. *bli-3* [14] and *mlt-7* [11] mRNA expression levels, encoding enzymes involved in the dityrosine crosslinking of worm collagen, were also tested. As shown in Figure 2B, B_12_ deficiency hardly affected *bli-3* and *mlt-7* mRNA expression levels compared with those of the control. These results suggest that the dityrosine crosslinking of collagen is triggered by B_12_ deficiency-induced oxidative stress.

### 2.3. Effect of Collagenase Treatment on the Cuticular Extracellular Matrix Epidermal Collagen Layer in Control and B_12_-Deficient Worms

To evaluate the effect of the highly formed dityrosine crosslinking of collagen, the epidermal collagen layer was treated with collagenase. When control and B_12_-deficient worms were treated with collagenase solution for 10 min, the epidermal collagen layer was mostly digested in the control worms but remained unaffected in the B_12_-deficient worms (Figure 3). When B_12_-deficient worms were grown for three generations under B_12_-supplemented conditions (recovery), the epidermal collagen layer of the recovery worms was readily digested by the collagenase treatment. These results indicate that the epidermal collagen layer of B_12_-deficient worms became collagenase-resistant due to the high-level dityrosine crosslinking, implying that cuticular collagen would become structurally stronger and more rigid in the B_12_-deficient worms compared with the control worms.

### 2.4. Effect of B_12_ Deficiency on C. elegans Motility Function

The B_12_ deficiency-induced high-level dityrosine crosslinking might potentially affect the physiological functions of the cuticular extracellular matrix. When the whiplash movement in the M9 buffer was evaluated as a motility function, the movement for 30 s in the B_12_-deficient worms (approximately 99.3 thrashes/30 s) was decreased up to approximately 80% of that of the control worms (approximately 125.9 thrashes/30 s) (Figure 4). The decreased motility function of B_12_-deficient worms was completely recovered to the control level when grown for three generations under B_12_-supplemented conditions (recovery). In addition, the AsA supplementation of B_12_-deficient worms showed that the reduced motility function was almost recovered to the control level. These results show that B_12_ deficiency leads to worm motility dysfunction due to decreased collagen level and increased dityrosine crosslinking formation in collagen, potentially reducing in flexibility of cuticular extracellular matrix.

### 2.5. Effect of B_12_ Deficiency on the Dityrosince Crosslinking Level of Aβ Peptides in GMC101 Worms

Dityrosine crosslinks generated by reactive oxygen species are reportedly formed in Aβ oligomers, and these dityrosine links further stabilize the fibrils in developing AD [13]. To elucidate the relationship between B_12_ deficiency and AD, we investigated whether reactive oxygen species induced by the B_12_ deficiency significantly increased the dityrosine crosslinking levels of Aβ peptides using *C. elegans* GMC101 mutant worms producing Aβ_1–42_ peptides in their muscle cells [15]. GMC101 mutant worms were grown until the young adult stage, under control and B_12_-deficient conditions; then, they were shifted from 20 °C to 25 °C to induce Aβ [15]. Half of the GMC101 worms grown under B_12_-deficient and control conditions showed signs of paralysis 48 and 96 h after the induction of Aβ, respectively (Figure 5). These results indicate that B_12_ deficiency significantly promoted Aβ-induced paralysis in GMC101 mutant worms.

To evaluate how B_12_ deficiency affects the Aβ mRNA and protein levels in GMC101 mutant worms, quantitative PCR and Western blot analyses were conducted. No significant changes could be observed between the control and B_12_-deficient mutant worm Aβ mRNA levels (Figure 6A). Figure 6B shows the protein expression levels in N2 wild-type and GMC101 mutant worms grown under B_12_-deficient and B_12_-supplemented conditions. Human Aβ antibody-immunoreactive components (A–F) were detected as oligomeric Aβ forms (approximately 17–31 kDa) only in GMC101 mutant worms, and monomeric Aβ (5 kDa) was not detected. An immunoreactive component with the molecular mass of 34 kDa was a non-specific component as the component was found in N2 wild-type worms. As shown in Figure 6C, no significant difference in the immunoreactive component A–F could be detected between the B_12_-supplemented and B_12_-deficient GMC101 mutant worms.

However, B_12_-deficient GMC101 mutant worms appear to accumulate slightly more Aβ in the pharynx and tail compared with control mutant worms (Figure 6D), although quantitative differences cannot be shown. These results indicate that B_12_ deficiency itself does not increase Aβ mRNA and protein expression levels.

To elucidate the effect of B_12_ deficiency on the dityrosine crosslinking level of Aβ in GMC101 mutant worms, dityrosine crosslinking was investigated using a fluorescent microscope. The dityrosine crosslinking of Aβ was detected in the whole body of B_12_-deficient mutant, but not in control mutant worms (Figure 7). The significant dityrosine crosslinking found in B_12_-deficient mutant worms considerably decreased when grown for three generations under B_12_-supplemented conditions (Recovery). Moreover, the dityrosine crosslinking of Aβ could be hardly found in AsA-supplemented B_12_-deficient mutant worms.

To clarify whether B_12_ deficiency-induced oxidative stress could promote the dityrosine crosslinking of Aβ, MDA and H_2_O_2_ levels were determined in GMC101 mutant worms grown under control, B_12_-deficient, recovery, and AsA-supplemented B_12_-deficient conditions (Figure 8A,B). The MDA and H_2_O_2_ levels significantly increased during B_12_ deficiency. The increased MDA and H_2_O_2_ levels of B_12_-deficient mutant worms were completely recovered to the control level when grown for three generations under B_12_-supplemented conditions (Recovery). Moreover, AsA supplementation did not show any increase in MDA and H_2_O_2_ levels even in B_12_-deficient mutant worms. To evaluate the effects of B_12_ deficiency on mRNA levels, encoding enzymes involved in cellular antioxidant systems, the superoxide dismutase (*sod-1*), and catalase (*ctl-1*) mRNA levels were measured by qPCR (Figure 8C). No significant changes in the *sod-1* and *ctl-1* mRNA levels were observed during B_12_ deficiency. Although B_12_ deficiency did not affect the mRNA levels of these antioxidant enzymes, it significantly increased the MDA and H_2_O_2_ levels in GMC101 mutant worms, suggesting that superoxide dismutase and catalase activities decreased due to the oxidative inactivation of the enzymes, as described in B_12_-deficient N2 wild-type worms [16]. These observations indicate that B_12_ deficiency-induced oxidative stress promoted the dityrosine crosslinking of Aβ.

## 3. Discussion

We previously reported that certain B_12_-deficient worms showed a short and plump phenotype, like “dumpy” mutants [7] formed due to the disordered cuticle collagen biosynthesis [17]. *C. elegans* possesses an external structure known as the cuticle, containing collagen and collagen-like protein major components of approximately 80% of the total cuticular protein content [18]. The cuticle is required to maintain body shape [19,20,21] and is synthesized five times from late embryogenesis throughout the *C. elegans* lifecycle, since the worm requires a new cuticle after molting in each growth stage [18]. Therefore, these observations indicate that the cuticle is crucial for the development and survival of worms. Worm cuticle collagen is synthesized and maturated by the following steps. Synthesized collagen single polypeptides are modified with proline hydroxylation and disulfide bond formation by the proline hydroxylase complex [DPY-18 (*dpy-18*), PHY-2 (*phy-2*), and PDI (*pdi-2*)] in the rough endoplasmic reticulum. After the secretion of the modified collagen triple helices into the extracellular space, they undergo a final modification of an enzymatic intermolecular tyrosine crosslinking by a dual oxidase BLI-3 (*bli-3*) [11]. Although a *C. elegans* gene ortholog (*duox-2*) of the human dual oxidase 2, involved in collagen intermolecular crosslinking, has been identified [22], *duox-2* was not expressed in the worms [22,23], as this gene might be a pseudo-gene. Furthermore, a heme peroxidase MLT-7 (*mlt-7*), along with BLI-3, reportedly played an essential role in cuticle–collagen crosslinking [11]. In contrast, the occurrence of tyrosine crosslinking is rare in vertebrates, and lysyl oxidase-derived linkages are predominant, while lysyl crosslinking is absent in the *C. elegans* cuticle [11].

As shown in Figure 1B and Figure 2B, B_12_ deficiency did not affect the mRNA expression levels of the proline hydroxylase complex (DPY-18, PHY-2, and PDI), dual oxidase (BLI-3), and heme peroxidase (MLT-7) involved in the posttranslational modification of worm collagen. However, worm collagen level was significantly decreased to approximately 59% of that in the control during B_12_ deficiency (Figure 1A). The decreased collagen level of B_12_-deficient worms was due to the significant decrease in AsA (approximately 52% of that of the control worms) (Figure 1C) as a coenzyme of prolyl 4-hydroxylase.

During B_12_ deficiency, homocysteine (Hcy) was significantly accumulated in the worm body [7]; leading to the disruption of redox regulation [16] due to severe Hcy-related oxidative stress [24]. Although B_12_ deficiency did not affect the mRNA expression levels of superoxide dismutase (*sod-1*) and catalase (*ctl-1*) involved in the cellular oxidant defense systems, the activities of superoxide dismutase and catalase were significantly reduced due to the oxidative inactivation of these enzymes [16]. Therefore, cellular antioxidant compound levels, such as those of glutathione and AsA were significantly reduced [16]. These observations indicate that AsA was mainly used as an antioxidant to scavenge oxidative stress induced during B_12_ deficiency and, consequently, was significantly decreased. Therefore, the decreased AsA induced decreased collagen biosynthesis in *C. elegans*. AsA deficiency has reportedly interfered with collagen synthesis in guinea pigs [25]. Moreover, it has been reported that Hcy, itself, can disrupt the collagen posttranslational modification in mammalian bones [10,26,27]. 

As shown in Figure 2A, dityrosine levels (per mg collagen) were significantly increased in the B_12_-deficient worms compared with the control worms, indicating that B_12_ deficiency significantly increases the crosslinking level of tyrosine residues in collagen. Tyrosine crosslinking was formed by reactive oxygen species, induced by significantly increased Hcy during B_12_ deficiency, as B_12_ deficiency hardly stimulated the mRNA expression levels of dual oxidase BLI-3 (*bli-3*) and the heme peroxidase MLT-7 (*mlt-7*) involved in the tyrosine crosslinking of the cuticular collagen. Lévigne et al. [28] have reported that a deficiency of certain NADPH oxidase involved in the production of reactive oxygen species reduced dityrosine crosslinking, coinciding with the results of this study. Severe oxidative stress caused by aging has reportedly increased the dityrosine crosslinking of worm collagen [11].

The above-presented results suggest that the “dumpy” mutants formed by B_12_ deficiency were due to reduced intracellular collagen biosynthesis. Furthermore, B_12_ deficiency significantly increased the formation of the dityrosine crosslinking of collagen as an extracellular maturation step. As shown in Figure 4, B_12_ deficiency leads to worm motility dysfunction, probably due to such disordered collagen biosynthesis and maturation, themselves being probably due to the reduction of cuticular extracellular matrix flexibility. Although thrashing rates generally reflect worm body wall muscles, we have no information available on whether B_12_ deficiency-induced structural or functional muscular disorders.

Although little information is available on the mechanism by which B_12_ deficiency. contributes to AD pathogenesis, numerous studies have reported that the serum Hcy, elevated by B_12_ deficiency, is associated with AD pathogenesis [29,30,31]. In this study, dityrosine crosslinking of the cuticular collagen was formed by B_12_ deficiency-induced oxidative stress. Similarly, oxidative stress due to B_12_ deficiency can induce the dityrosine crosslinking of Aβ peptides in developing AD, as Aβ oligomers are formed by the dityrosine crosslinking generated by reactive oxygen species [13]. In GMC101 mutant worms, the MDA and H_2_O_2_ levels significantly increased during B_12_ deficiency, although no significant changes could be observed in the mRNA levels of the superoxide dismutase (*sod-1*) and catalase (*ctl-1*) involved in cellular oxidant defense systems. Similar results were reported in B_12_-deficient N2 wild-type worms exhibiting oxidative inactivation of superoxide dismutase and catalase, leading to the disruption of redox regulation [16]. Therefore, we evaluated whether oxidative stress generated by B_12_ deficiency can form toxic Aβ oligomers using a GMC101 mutant worm that expresses full-length human Aβ peptide in the muscle cells. When Aβ oligomerization is formed and its toxicity is induced, mutant worms have shown paralysis [15]. Therefore, the GMC101 mutant worm has been used as a model animal of AD [32]. Our previous study [33] indicated that B_12_ deficiency did not stimulate the production of Aβ peptides in GMC101 mutant worms, although B_12_-deficient worms exhibited paralysis faster and more severely than B_12_-sufficient worms (control) did. As shown in Figure 5, similar results were obtained in this study. Furthermore, AsA-supplemented B_12_-deficient worms rescued the paralysis phenotype. However, AsA supplementation did not affect Aβ peptide aggregations, suggesting that oxidative stress caused by elevated Hcy levels is an important factor in toxicity.

As shown in Figure 7, the dityrosine crosslinking of Aβ was present in the entire bodies of B_12_-deficient mutant worms, but not in those of control mutant worms. The dityrosine crosslinking of Aβ could hardly be found in AsA-supplemented B_12_-deficient mutant worms. As shown in Figure 6, no significant changes could be observed in Aβ oligomerization between the control and B_12_-deficient mutant worms. However, the B_12_-deficient worms exhibited paralysis due to toxicity faster than control worms did (Figure 5). Sitkiewicz et al. [34] demonstrated that tyrosine crosslinking shifts the equilibrium toward more compact oligomer types, leading to highly toxic fibrils. Maina et al. [35] suggest that dityrosine crosslinking, specifically, promotes the stabilization, but not the induction or facilitation, of Aβ assembly, and Aβ exerts high-level toxicity at a stage when self-assembly is high. These observations and the results presented in worms suggest that the dityrosine crosslinking formed during B_12_ deficiency promotes the formation and stabilization of the compact Aβ oligomers that facilitate self-assembly to induce high toxicity, probably in neuronal cells specifically located in the pharynx and tail (Figure 6D).

## 4. Materials and Methods

### 4.1. Organisms

The N2 Bristol wild-type *C. elegans* strain was maintained at 20 °C on nematode growth medium (NGM) plates using the *Escherichia coli* OP50 strain as a food source [36]. B_12_-supplemented (control) and B_12_-deficient worms were prepared as previously described [7]. B_12_-deficient worms were transferred to a B_12_-supplemented medium for three generations and used as the recovery worms. In the case of *L*-ascorbic acid (AsA)-supplemented experiments, B_12_-deficient worms were grown in a B_12_-deficient medium containing AsA 2-glucoside (final concentration of 1 mM) for three generations [16]. The transgenic GMC101 mutant worm strain was obtained from the Caenorhabditis Genetics Center (University of Minnesota, Minneapolis, MN, USA). The mutant worms were backcrossed five times before experimental use. When the mutant worms developed into young adults, their cultivation temperature was shifted from 20 °C to 25 °C to induce Aβ. B_12_ deficiency was also induced in the case of the GMC101 mutant worms through the above-described method. All nematodes were synchronized to obtain an identical developmental stage for experimental use.

### 4.2. Worm Body Collagen Determination

The acid hydrolysis of worm-body proteins was conducted according to the modified method of Roach and Gehrke [37]. Briefly, the worms (approximately 0.05 g wet weight of each background) were homogenized in 500 μL of 6-M HCl using a hand homogenizer (AS ONE Corp., Osaka, Japan). The homogenates were transferred into glass reaction tubes and supplemented with 6-M HCl (500 μL), and the pressure of the reaction tubes was reduced. After the homogenates were hydrolyzed under reduced pressure at 110 °C for 24 h, the resulting hydrolysates were centrifuged at 15,000× *g* for 10 min at 4 °C. Each supernatant (250 μL) was diluted with an equal volume of 0.25 mol/L lithium citrate buffer (pH 2.2) (Fujifilm Wako Pure Chemical, Osaka, Japan) and filtered using a Millex^®^-LH membrane filter (Merck Millipore, Darmstadt, Germany). Hydroxyproline was analyzed using a fully automated amino acid analyzer (JEOL JLC-500/V, Nihon Denshi Datem Corp. Ltd., Tokyo, Japan). The worm collagen content was calculated from the determined hydroxyproline values using a conversion factor of 8.33 [19].

### 4.3. AsA Determination

The worms (approximately 0.05 g wet weight of each background) were homogenized in 300 μL of 5% (*w*/*v*) metaphosphoric acid solution on ice using a hand homogenizer (AS ONE). After the homogenates were centrifuged at 15,000× *g* for 10 min at 4 °C, the supernatants were used as samples. AsA was assayed according to the 2,4-dinitrophenyl hydrazine derivatization method [38]. Briefly, after AsA was completely oxidized to dehydro-form using indophenol solution, the formed dehydroAsA was derivatized with 2,4-dinitrophenyl hydrazine to form its osazone. The formed derivative compound was determined using a Shimadzu High-Performance Liquid Chromatography (HPLC) system (SPD-6AV UV-VIS Spectrophotometric Detector, LP-6A Liquid Delivery Pump, and CTO-6V Column Oven) with a CDS ver. 5 chromato-data processing system (LAsoft, Ltd., Chiba, Japan). Each sample (20 μL) was applied onto a Normal Phase HPLC Column (Senshu Pak Silica-2150-N, φ 6.0 × 150 mm, Senshu Scientific Corp. Ltd., Tokyo, Japan) and eluted with acetic acid/hexane/ethyl acetate (1:4:5, *v*/*v*/*v*) as a mobile phase at 40 °C. The flow rate was 1.5 mL/min. The derivative compound was monitored by measuring the absorbance at 495 nm.

### 4.4. Dityrosine Determination

Dityrosine was determined using the Shimadzu HPLC system (PU-2080 Plus Intelligent HPLC Pump, DG-2080-53 Degasser, RF-530 Fluorescence HPLC Monitor, and CTO-20A Column Oven) according to the method of Thein et al. [11]. Briefly, each hydrolyzed worm protein sample (20 µL), prepared as described above, was loaded onto a reversed-phase HPLC column (Luna C18 (2) 5 µm, 250 mm × 4.6 mm 100 Å; Phenomenex., Torrance, CA, USA) and isocratically eluted with 0.1-M KH_2_PO_4_-phosphoric acid (pH 3.8) as a mobile phase, at 40 °C. The flow rate was 1 mL/min. Dityrosine was monitored by measuring the fluorescence with an excitation and emission at 285 and 410 nm, respectively.

### 4.5. Collagenase Treatment

Worms (approximately 100 individuals) grown under various conditions were washed three times with M9 buffer (3 g KH_2_PO_4_, 6 g Na_2_HPO_4_, 0.5 g NaCl, and 1 g NH_4_Cl/L) and then fixed with 4% (*w*/*v*) paraformaldehyde for 10 min at 4 °C. The fixed worms were washed three times with phosphate-buffered saline (PBS) buffer (pH 7.2). After the worms were soaked with 1.5 mL of β-mercaptoethanol buffer containing β-mercaptoethanol (75 μL), distilled water (1222.5 μL), 1 M Tris-HCl (pH 6.9) (187.5 μL), and Triton X-100 (15 μL) for 30 min using a rotator, they were washed three times with PBS buffer (pH 7.2). The washed worms were treated with 450 μL of collagenase solution (1 unit/μL) for 10 min at room temperature (25 °C), and the reaction vessel was placed on ice for 30 min to stop the enzyme reaction. After the treated worms were immediately washed three times with PBS buffer (pH 7.2), each worm was mounted on glass slides. The C. elegans cuticular collagen layer was observed under an ECLIPSE Ts2 microscope (Nikon Corp., Tokyo, Japan).

### 4.6. Assays of Malondialdehyde and H_2_O_2_ as Oxidative Stress Markers

Worms (approximately 0.05 g wet weight of each background) grown under various conditions were homogenized in 200 μL of 100 mM potassium-phosphate buffer (pH 7.0) on ice using a hand homogenizer (AS ONE). After the homogenates were centrifuged at 15,000× *g* for 10 min at 4 °C, the supernatants were used as samples. Malondialdehyde (MDA) and H_2_O_2_ were determined using a TBARS assay kit (ZeptoMetrix Corp., Buffalo, NY, USA) and a H_2_O_2_ assay kit (BioVision, Inc., Milpitas, CA, USA), respectively. The MDA–thiobarbituric acid adducts or the reaction product of the OxiRed probe and H_2_O_2_ in the presence of horseradish peroxidase formed in the samples were determined by measuring the absorbance at 540 or 570 nm, respectively, using a Sunrise Rainbow RC-R microplate reader (Tecan Austria GmbH, Salzburg, Austria).

### 4.7. Immunofluorescent Staining of Aβ and Dityrosine in GMC 101 Mutant Worms

For visualizing Aβ peptides and their dityrosine crosslinking in the worm body, worms (approximately 100 individuals), grown under veracious conditions, were washed three times with M9 buffer and then fixed using 4% (*w*/*v*) paraformaldehyde for 10 min at 4 °C. The fixed worms were washed three times with PBS buffer (pH 7.2). After the worms were soaked with 1.5 mL of β-mercaptoethanol buffer, as described above, for 30 min using a rotator, they were washed three times with PBS buffer (pH 7.2). The washed worms were treated with 450 μL of collagenase solution (1 unit/μL) for 13 or 15 min at room temperature (25 °C), and the reaction vessel was placed on ice for 30 min to stop the enzyme reaction. The treated worms were immediately washed three times with PBS buffer (pH 7.2) and then were treated with 500 μL of blocking solution [5 mg bovine serum albumin (BSA) and 5 μL Triton X-100 per 1 mL of PBS (pH 7.2)] for 30 min at room temperature (25 °C) to block nonspecific antibody binding. The treated worms were washed three times with a washing buffer [3 mg BSA and 5 μL Triton X-100 per 1 mL of PBS (pH 7.2)]. After the worms were treated with an anti-β amyloid 1-42 rabbit monoclonal antibody (ab180956, Abcam, Cambridge, MA, USA) or an anti-dityrosine mouse monoclonal antibody (Nikken Seil Co., Ltd., Shizuoka, Japan) for 24 h at room temperature (25 °C), they were washed with the abovementioned washing buffer. The worms were treated with an anti-rabbit IgG secondary antibody (20-fold dilution) (ab6717, Abcam) or an anti-mouse IgG secondary antibody (20-fold dilution) (ab6785, Abcam) coupled to fluorescein isothiocyanate under dark conditions for 1 h. Next, the worms were washed with the above-described washing buffer and mounted on glass slides. Aβ and dityrosine visualization in the worms was performed using an ECLIPSE Ts2 fluorescent microscope (Nikon Corp.).

Immunoblot analysis was performed as previously described [33]. Sodium dodecyl sulfate polyacrylamide gel electrophoresis was performed using p-PAGEL slab gels (P-T16.5S; ATTO Corp., Tokyo, Japan). Aβ peptide was detected using a monoclonal anti-Aβ_1–42_ primary antibody (EPR9296, Abcam) and anti-rabbit IgG-horseradish peroxidase conjugate (ab6721, Abcam). Signals were detected using EzWestBlue (ATTO Corp.) according to the manufacturer’s instructions. The chemical coloring intensity was quantified using ImageJ (ImageJ Software, Bethesda, MD, USA) for three independent experiments.

### 4.8. Quantitative Polymerase Chain Reaction (qPCR) Analysis

Worm total RNA was prepared using Sephasol^®^-RNA1 (Nacalai Tesque Inc., Kyoto, Japan). Poly(A)+ mRNA was prepared from the total RNA using the Poly (A)+ Isolation Kit from Total RNA (Nippon Gene, Tokyo, Japan) and then was used to synthesize cDNA using a PrimeScript™ II 1st Strand cDNA Synthesis Kit (Takara Bio, Otsu, Japan). The primer pairs used for the qPCR analysis were designed using the GENETYX software (GENETYX Corp., Tokyo, Japan) to yield 20–24-nucleotide sequences with approximately 100–150-bp amplification products. A CFX Connect™ Real-Time System (Bio-Rad) with SYBR Premix Ex Taq (Takara Bio) was used to perform qPCR. β-Actin (*act-1*) was used as an internal standard. The qPCR experiments were repeated at least three times for each cDNA prepared from three preparations of worms. Table 1 shows the primers used for qPCR.

### 4.9. Swim Locomotion Analysis

Worm motility function was evaluated using the swim locomotion method as follows. Individual three-day-old worms were placed on NGM agar plates (diameter of 3 cm) filled with 1.5 mL M9 buffer; then, their swimming motion was video-recorded for 1 min using a microscope (ECLIPSE Ts2, Nikon Corp.) equipped with a video system (DS-Fi3 camera unit and DS-L4 DS camera controller unit, Nikon Corp.). One round trip of worm head in the M9 buffer was determined as one whiplash movement.

### 4.10. Paralysis Assay

GMC101 mutant worms were grown at 25 °C after the L4 stage for Aβ induction. Worms that could move their heads but failed to move their bodies were scored as paralyzed [39]. The paralysis assay was performed every 12 h using approximately 50 individual worms.

### 4.11. Statistical Analysis

The results shown in Figure 1A,C, Figure 2A, Figure 4 and Figure 8 were analyzed using one-way ANOVA with Bonferroni’s post hoc test using GraphPad Prism 4 (GraphPad Software, La Jolla, CA, USA). The results shown in Figure 1B, Figure 2B, Figure 5 and Figure 6A,C were analyzed using Student’s *t*-test for pairwise comparison. All data, except for those presented in Figure 5, are presented as the mean ± SEM. Differences were considered statistically significant at *p* < 0.05.

## Figures and Tables

**Figure 1 ijms-22-12959-f001:**
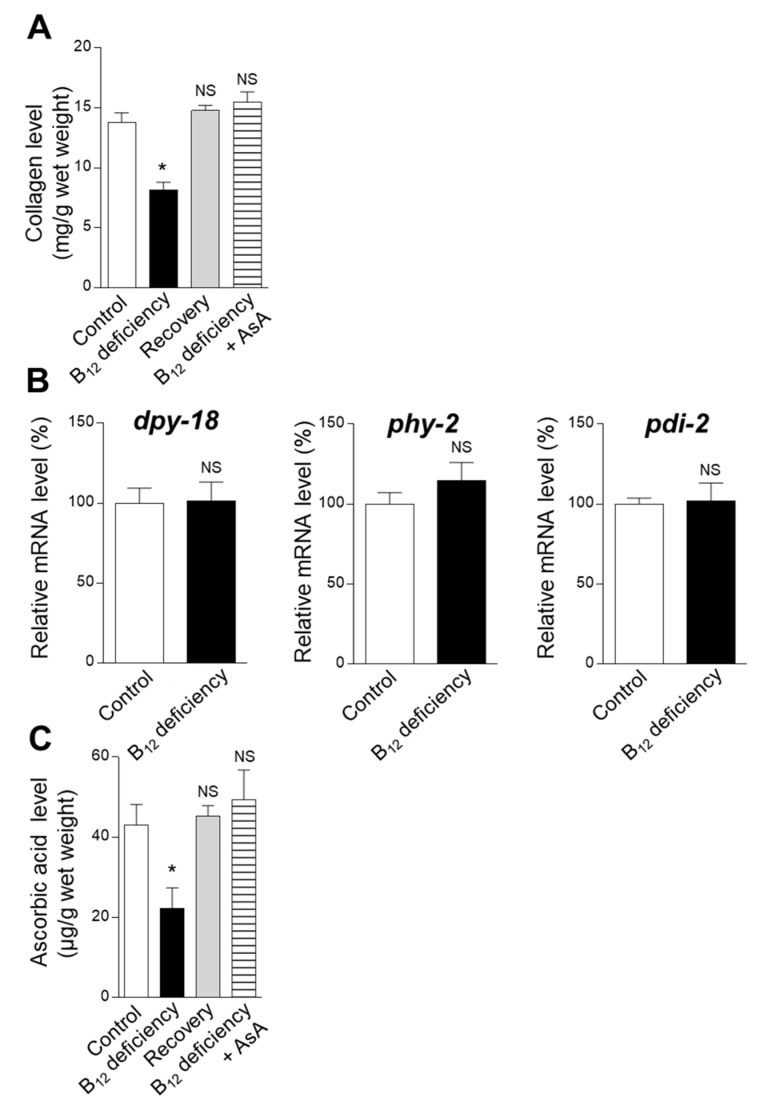
Effects of collagen biosynthesis during B_12_ deficiency in *C. elegans*. (**A**) Collagen levels calculated from the hydroxyproline content, (**B**) the mRNA expression levels of genes encoding complex prolyl 4-hydroxylase proteins, and (**C**) vitamin C (AsA) levels determined in B_12_-supplemented worms (Control), B_12_-deficient worms (B_12_ deficiency), B_12_-deficient worms grown for three generations under B_12_-supplemented conditions (Recovery), and B_12_-deficient worms grown for three generations under an AsA-supplemented condition (B_12_ deficiency + AsA). Shown are the mRNA expression levels of human prolyl 4-hydroxylase subunit α1 (P4HA1) and prolyl 4-hydroxylase subunit α2 (P4HA2), and of prolyl 4-hydroxylase subunit β (P4HB) and their genetic orthologs *dpy-18*, *phy-2*, and *pdi-2*, respectively. The data represent the mean ± SEM of three independent experiments (n = 3). * *p* < 0.05 versus the control group. NS represents no significant differences.

**Figure 2 ijms-22-12959-f002:**
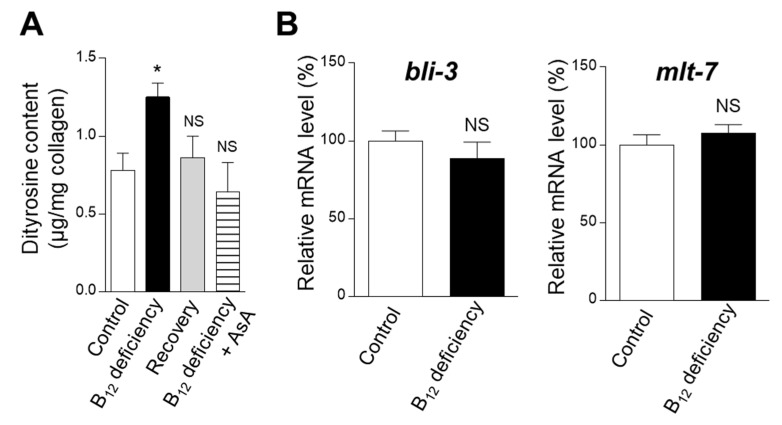
Effects of dityrosine and mRNA expression levels of enzymes involved in dityrosine crosslinking during B_12_ deficiency in *C. elegans*. (**A**) Dityrosine (per 1 mg collagen) and (**B**) mRNA expression levels (*bli-3* and *mlt-7*) determined in B_12_-supplemented worms (Control), B_12_-deficient worms (B_12_ deficiency), B_12_-deficient worms grown for three generations under B_12_-supplemented conditions (Recovery), and B_12_-deficient worms grown for three generations under AsA-supplemented conditions (B_12_ deficiency + AsA). The data represent the mean ± SEM of three independent experiments (n = 3). * *p* < 0.05 versus the control group. NS represents no significant differences.

**Figure 3 ijms-22-12959-f003:**
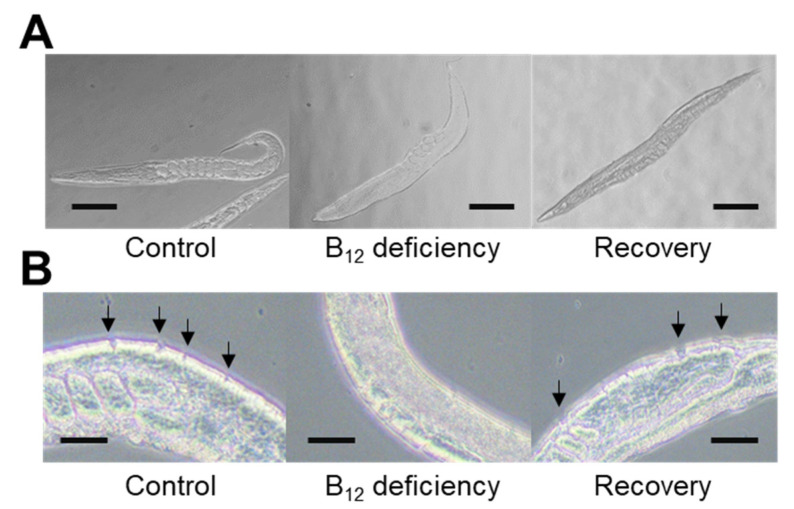
The morphological changes during B_12_ deficiency and collagen layer states after collagenase treatment. (**A**) Each worm (approximately 100 individuals) was washed three times with M9 buffer (3 g KH_2_PO_4_, 6 g Na_2_HPO_4_, 0.5 g NaCl, and 1 g NH_4_Cl/L) and then imaged using a microscope system. Scale bars = 250 µm. (**B**) Each worm (approximately 100 individuals) was washed three times with the same M9 buffer and then fixed using 4% (*w*/*v*) paraformaldehyde for 10 min at 4 °C. The fixed worms were washed three times with PBS buffer (pH 7.2). Subsequently, the worms were soaked with 1.5 mL of β-mercaptoethanol buffer for 30 min using a rotator; then, the worms were washed three times with PBS buffer (pH 7.2). The washed worms were treated with collagenase solution for 10 min at room temperature and then kept on ice for 30 min to stop the reaction. Immediately after that, the worms were washed three times with PBS buffer (pH 7.2); then, each worm was mounted on glass slides. A microscope was used to observe the collagen layer. The arrow (↓) indicates the collagen layer decomposed by the collagenase treatment. Scale bars = 25 µm. Control, B_12_ deficiency, and Recovery represent B_12_-supplemented worms, B_12_-deficient worms, and B_12_-deficient worms grown for three generations under B_12_-supplemented conditions, respectively.

**Figure 4 ijms-22-12959-f004:**
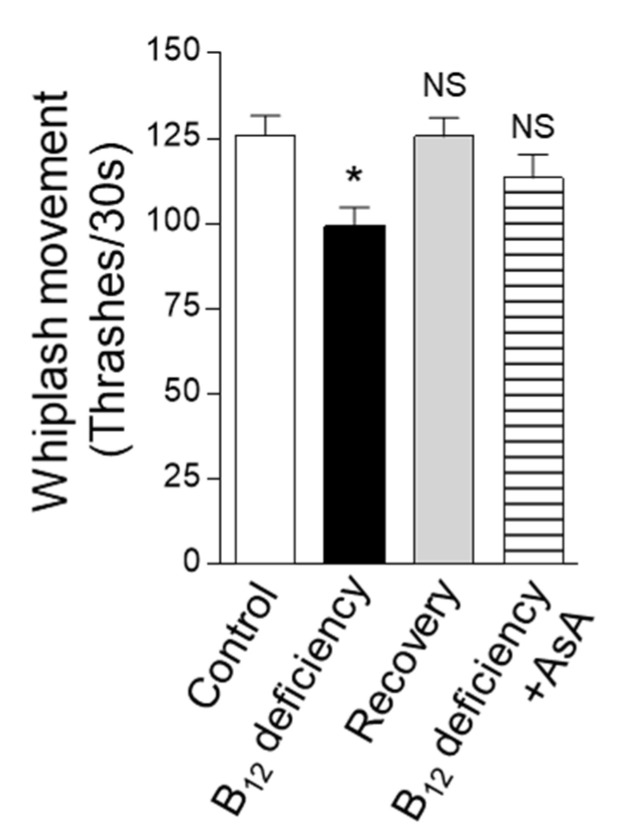
Effects of B_12_ deficiency on *C. elegans* motility. Whiplash movement with underwater condition was determined in B_12_-supplemented (Control), B_12_-deficient (B_12_ deficiency), B_12_-deficient worms grown for three generations under B_12_-supplemented conditions (Recovery), and B_12_-deficient worms grown for three generations under AsA-supplemented conditions (B_12_ deficiency + AsA). The data represent the mean ± SEM of whiplash movement of 50 individual animals. * *p* < 0.05 versus the control group. NS represents no significant differences.

**Figure 5 ijms-22-12959-f005:**
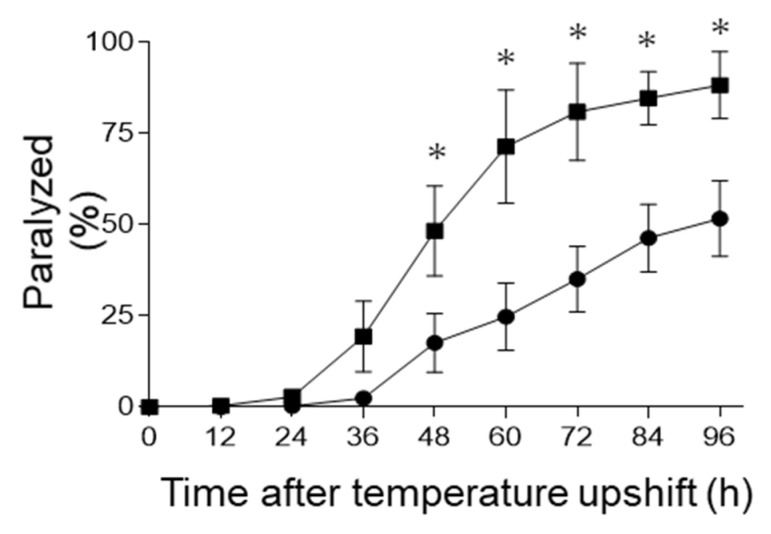
Paralysis rate of B_12_-supplemented and B_12_-deficienct GMC101 mutants. When GMC101 mutants grown under B_12_-supplemented (●) and B_12_-deficient conditions (■) developed into young adults, each GMC101 strain was shifted from 20 °C to 25 °C for the Aβ induction. The mean percentage of paralyzed worms is plotted against the time post temperature shift (h). All values represent the mean ± SD of five independent experiments (N = 5). Approximately 250 worms were screened for each condition. Asterisks indicate significant differences compared with the B_12_-supplemented worms at the same time point (* *p* < 0.05).

**Figure 6 ijms-22-12959-f006:**
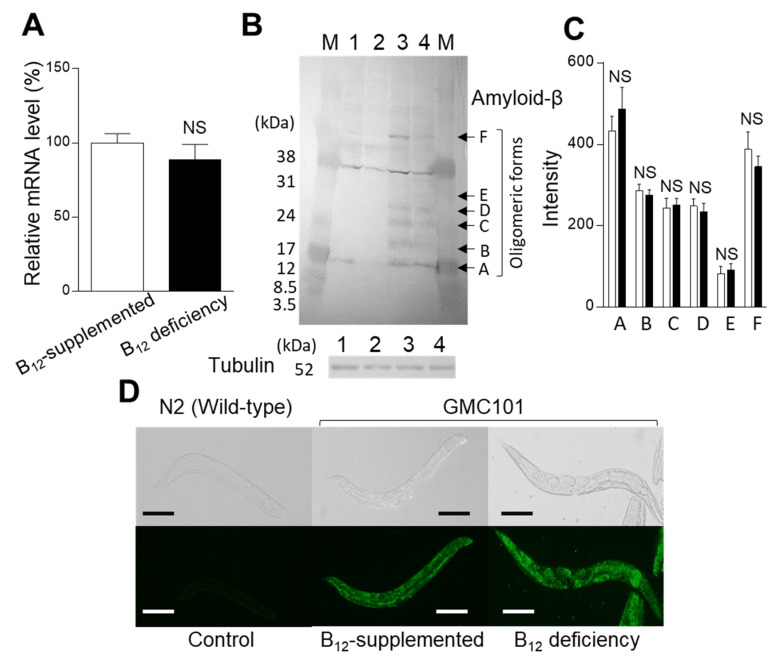
Effects of B_12_ deficiency on the Aβ mRNA and protein expression levels in GMC101 mutants. (**A**) mRNA expression levels in B_12_-supplemented GMC101 mutant (B_12_-supplemented) and B_12_-deficient GMC101 mutant worms (B_12_ deficiency). The data represent the mean ± SEM of three independent experiments. NS: no significant differences. (**B**) protein expression levels in B_12_-supplemented N2 wild-type worms (1), B_12_-deficient N2 wild-type worms (2), B_12_-supplemented GMC101 mutants (3), and B_12_-deficient GMC101 mutants (4). M: molecular mass marker proteins. A, B, C, D, E, and F: human Aβanti-body-immunoreactive components (oligomeric Aβ forms). (**C**) Relative amounts of oligomeric Aβ forms in B_12_-supplemented (white bar) and B_12_-deficient (black bar) GMC101 mutants. Immunoreactive components A-F, detected in panel B, were quantified using the ImageJ software. The data represent the mean ± SEM of three independent experiments. NS: no significant differences. (**D**) Fluorescent images of Aβ in N2 wild-type and GMC101 mutant animals. B_12_-supplemented N2 (Control), B_12_-supplemented GMC101 mutants (B_12_-supplemented), and B_12_-deficient GMC101 mutants (B_12_ deficiency). Scale bars = 200 µm.

**Figure 7 ijms-22-12959-f007:**
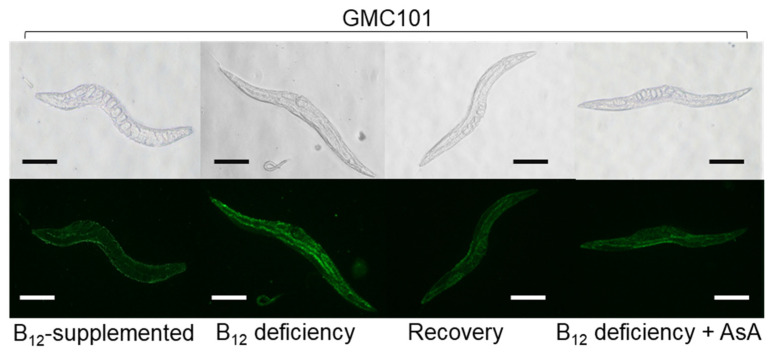
Dityrosine crosslinking accumulation in GMC101 mutants during B_12_ deficiency. Fluorescent images of dityrosine in GMC101 mutants in B_12_-supplemented mutants (B_12_-supplemented), B_12_-deficient mutants (B_12_ deficiency), B_12_-deficient mutants grown for three generations under B_12_-supplemented conditions (Recovery), and B_12_-deficient mutants grown in AsA-supplemented medium for three generations (B_12_ deficiency + AsA). Scale bars = 200 µm.

**Figure 8 ijms-22-12959-f008:**
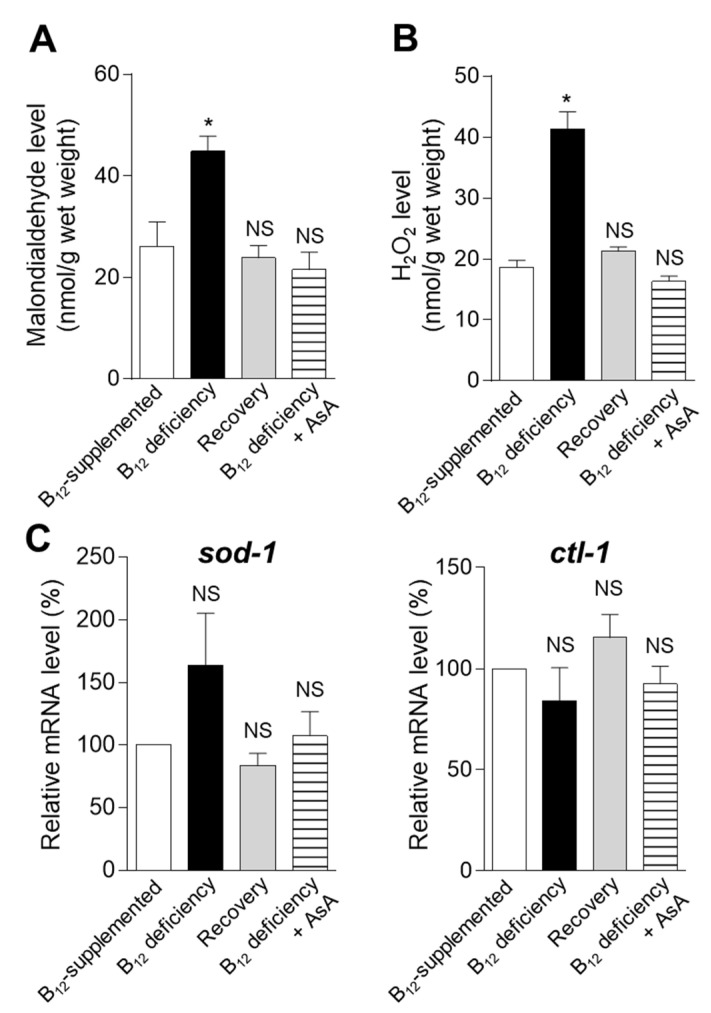
Effect of B_12_ deficiency on oxidative stress marker and oxidative defense enzyme levels in GMC101 mutants. (**A**) MDA and (**B**) H_2_O_2_ levels in GMC101 mutants are shown in B_12_-supplemented mutants (B_12_-supplemented), B_12_-deficient mutants (B_12_ deficiency), B_12_-deficient mutants grown for three generations under B_12_-supplemented conditions (Recovery), and B_12_-deficient mutants grown in AsA-supplemented medium for three generations (B_12_ deficiency + AsA). (**C**) Levels of mRNAs encoding enzymes involved in oxidant defense enzymes in GMC101 mutant animals. Using qPCR, we determined the levels of mRNAs encoding superoxide dismutase (*sod-1*) and catalase (*ctl-1*). The data represent the mean ± SEM of three independent experiments. * *p* < 0.05 versus the control group. NS represents no significant differences.

**Table 1 ijms-22-12959-t001:** Primer pairs used for the qPCR analysis.

Genes	Sequence (5′–3′)
*dpy-18* (Sense)	CTACCACACTGTGATGTGGATG
*dpy-18* (Antisense)	GCGTGCTTCAAGTTGTTCTG
*phy-2* (Sense)	GCTTGATGTGTGGATGCAGGTT
*phy-2* (Antisense)	TTGCGAGTCGTTTGGTGAGA
*pdi-2* (Sense)	CGGAATCGATGATGTTCCATTCGG
*pdi-2* (Antisense)	TTGGGTGAGCTTCTCGTCGAAAG
*bli-3* (Sense)	GCGCTCAAAACATGTGCTGT
*bli-3* (Antisense)	GCCAGATTGTTGTACCATCCGT
*mlt-7* (Sense)	TTGCGATCATCACGAGTGGTGT
*mlt-7* (Antisense)	AGCAGTTGTCGTGACTGGCAAA
Aβ (Sense)	GCGGATGCAGAATTCCGACATGAC
Aβ (Antisense)	TATGACAACACCGCCCACCATGAG
*sod-1* (Sense)	TCTTCTCACTCAGGTCTCCAAC
*sod-1* (Antisense)	TCGGACTTCTGTGTGATCCA
*ctl-1* (Sense)	ATTATGCTCGTGGTGGAAACCC
*ctl-1* (Antisense)	ACAATGTTTGGCGCCCTCAA
*act-1* (Sense)	TCCAAGAGAGAGGTATCCTTACCC
*act-1* (Antisense)	CTCCATATCATCCCAGTTGGTG

The qPCR primer pairs were designed using the GENETYX software. For normalization, β-actin (*act-1*) served as the internal standard.

## Data Availability

Data sharing not applicable.

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
