# Peer review of "Dityrosine Crosslinking of Collagen and Amyloid-β Peptides Is Formed by Vitamin B12 Deficiency-Generated Oxidative Stress in Caenorhabditis elegans"

_ijms, 2021, doi:10.3390/ijms222312959_

Round 1
Reviewer 1 Report
The manuscript is an interesting paper and nicely done. However, authors are encouraged authors to provide the statistical result of Fig 6B.
Author Response
Dear Reviewer 1,
Thank you very much for your decision letter of 11th, November 2021, with regard to our manuscript (ijms-1455341) with the comments from yourself. We appreciate the comments, which are very helpful. We have tried to revise the manuscript in line with suggestions.
In response to comments from the reviewer 1, the following changes were made (as marked in blue).
- Authors are encouraged authors to provide the statistical result of Fig 6B
Ans: New Figure 6C, the legend of Figure 6, and the relating text have been added in the revised manuscript according to the Reviewer’s suggestion (Lines 190-201 and Lines 234-243).
I hope that you will be able to consider this manuscript for possible publication in International Journal of Molecular Sciences.
Thank you.
Sincerely,
Dr. Tomohiro Bito
Assistant professor
Department of Agricultural, Life and Environmental Sciences,
Faculty of Agriculture,
Tottori University
Reviewer 2 Report
In this manuscript the authors report that in C. elegans vitamin B12 deficiency results in decreased levels of collagen without affecting expression of the prolyl 4-hydroxylase enzyme involved in collagen biosynthesis. Protein levels of ascorbic acid, a prolyl 4-hydroxylase coenzyme, are instead decreased, suggesting that this may account for the decrease in collagen levels. Levels of dityrosine, which is involved in collagen crosslinking and extracellular maturation, are found to increase in B12 deficient worms, while mRNA expression levels of dityrosine crosslinking enzymes bli-3 and mlt-7 are not affected. In general this works adds to the list of developmental defects associated with vit B12 deficiency, and provides a plausible mechanism for the dumy phenotype observed in vit B12 deficient animals. However, the data provided does not support some of the main claims made, and the quality of figures could be improved.
Major comments:
In general the quality of the figures should be improved
- Throughout-labeling of X axis: It would make it easier for the reader if rather than numbering 1-4 specific conditions were written for each bar graph (wt, B12 deficiency etc..).
- Authors could provide better quality figures: orient animal in same direction-head to left/tail to right, increase contrast, provide images with better resolution (images appear out of focus). Could authors show a field with several animal for GFP reporter assays (figure 6 and 7).
- the authors use thrashing to measure motility; they show decreased thrashing in B12 defiient worms, and show rescue of this phenotype by B12 supplementation. However, the claim that decreased collagen levels and increased dityrosine crosslinking are responsible for the thrashing defect is not supported by the data. The authors previously showed that Vitamin B12 deficiency results in pleiotropic phenotypes, and thrashing rates generally reflect body wall muscle integrity.
Is there additional evidence to claim the defect is due to defects in cuticle structure?
- why does it take three generations for B12 supplementation to have an effect? is there no effect after 1 generation or two? do the authors have an explanation for this?
- Because levels of dityrosine crosslinking enzymes bli-3 and mlt-7 are not affected by B12 deficiency, the authors conclude that the observed dityrosine crosslinking of collagen is instead nonenzymatically triggered by B12 deficiency-induced oxidative stress. The authors also make this claim in the abstract, yet I see no experimental evidence to support this claim.
Figure 8 shows an increase in MDA levels (used as a marker of oxidative stress) in B12 deficient animals. Did the authors look at more direct indicators of oxidative stress? does the expression of any of the oxidative stress-induced genes (ctl-1, sod genes...?) correlate with vitamin B12 deficiency and complementation?
Even if these animals show more oxidative stress, none of the data presented shows that the scored phenotypes are the consequence of oxidative stress, as the authors imply.
- western blot in figure 6 should be labeled so reader knows what bands to look at. The tubulin control is very weak
- reference list should be updated and include more recent articles
- the introduction and discussion could be more clearly written
Author Response
Dear Reviewer 2,
Thank you very much for your decision letter of 11th, November 2021, with regard to our manuscript (ijms-1455341) with the comments from yourself. We appreciate the comments, which are very helpful. We have tried to revise the manuscript in line with suggestions.
In response to comments from the reviewer 2, the following changes were made (as marked in yellow).
- Throughout-labeling of X axis: It would make it easier for the reader if rather than numbering 1-4 specific conditions were written for each bar graph (wt, B12 deficiency etc..).
Ans: Labeling of X-axis in all Figures have been revised according to the reviewer’s suggestions.
- Authors could provide better quality figures: orient animal in same direction-head to left/tail to right, increase contrast, provide images with better resolution (images appear out of focus). Could authors show a field with several animal for GFP reporter assays (figure 6 and 7).
Ans: Figures 6 and 7 have been revised according to the reviewer’s suggestions. Unfortunately, GFP reporter assay cannot be done at the present time.
- the authors use thrashing to measure motility; they show decreased thrashing in B12 defiient worms, and show rescue of this phenotype by B12 However, the claim that decreased collagen levels and increased dityrosine crosslinking are responsible for the thrashing defect is not supported by the data. The authors previously showed that Vitamin B12 deficiency results in pleiotropic phenotypes, and thrashing rates generally reflect body wall muscle integrity. Is there additional evidence to claim the defect is due to defects in cuticle structure?
Ans: Figure 4 has been added in the data of B12-deficient worms grown under ascorbic acid-supplemented conditions (Lines 146-147). Unfortunately, at the present time we have no information available on cuticle structure or muscle integrity of B12-deficient worms. A sentence has been added at Lines 316-318 in revised manuscript.
- why does it take three generations for B12 supplementation to have an effect? is there no effect after 1 generation or two? do the authors have an explanation for this?
Ans: Unpublished data indicate that B12-treatment for three generations was required for complete recovery from the severe B12 deficient status to normal B12 (control) level.
- Because levels of dityrosine crosslinking enzymes bli-3 and mlt-7are not affected by B12 deficiency, the authors conclude that the observed dityrosine crosslinking of collagen is instead nonenzymatically triggered by B12 deficiency-induced oxidative stress. The authors also make this claim in the abstract, yet I see no experimental evidence to support this claim.
Figure 8 shows an increase in MDA levels (used as a marker of oxidative stress) in B12. deficient animals. Did the authors look at more direct indicators of oxidative stress? does the expression of any of the oxidative stress-induced genes (ctl-1, sod genes...?) correlate with vitamin B12 deficiency and complementation?
Even if these animals show more oxidative stress, none of the data presented shows that the scored phenotypes are the consequence of oxidative stress, as the authors imply.
Ans: As described previously [Bito etl al., Redox Biol, 2017, 11, 21-29], during B12 deficiency, homocysteine was significantly accumulated in the worm body; leading to the disruption of redox regulation due to severe Hcy-related oxidative stress. B12 deficiency did not affect the mRNA expression levels of superoxide dismutase (sod-1) and catalase (ctl-1) involved in cellular oxidant defense systems, the activities of superoxide dismutase and catalase were significantly decreased due to the oxidative inactivation of the enzymes. Therefore, cellular antioxidant compound levels, such as those of glutathione and ascorbic acid were significantly reduced. In addition, similar data that were obtained in GMC101 mutant have been given in Figure 8 B and C. Some sentences have been added at Lines 221-230, Lines 289-292, Lines 325-330, and Lines 435-441.
- western blot in figure 6 should be labeled so reader knows what bands to look at. The tubulin control is very weak
Ans: Figure 6B has been completely revised according to the reviewer’s suggestion.
- reference list should be updated and include more recent articles
Ans: We have tried to make the reference updated (Ref. No. 1, 2, 9, 18, 31, 32, and 34).
- the introduction and discussion could be more clearly written
Ans: The introduction and discussion sections have been revised according to the reviewer’s suggestion (Lines 56-64, Lines 259-261, Lines 289-292).
I hope that you will be able to consider this manuscript for possible publication in International Journal of Molecular Sciences.
Thank you.
Sincerely,
Dr. Tomohiro Bito
Assistant professor
Department of Agricultural, Life and Environmental Sciences,
Faculty of Agriculture,
Tottori University
Reviewer 3 Report
Authors carried out a series of experiments and showed in C. elegans that vitamin B12-deficiency affected motility dysfunction due to the decreased collagen level with highly tyrosine crosslinked collagen. They further showed that the same condition resulted in the di-tyrosine cross-links in Ab peptides, of which results were due to nonenzymatic processes. The experimental design and data interpretation are fine and I have only minor comments.
1) There are no information in Abstract on Abeta results. Authors need to add the major results on Ab in the Abstract.
2) It would help readers to grasp the message of figures if authors provide a legend inset representing the corresponding bars instead of numbering in x-axis in Fig. 1, 2 & 4.
3) line 135-136. Check the English grammar.
"These results indicated that the epidermal collagen layer of B12-deficient worms because collagenase-resistant due to the high-level dityrosine crosslinking;"
Author Response
Dear Reviewer 3,
Thank you very much for your decision letter of 11th, November 2021, with regard to our manuscript (ijms-1455341) with the comments from yourself. We appreciate the comments, which are very helpful. We have tried to revise the manuscript in line with suggestions.
In response to comments from the reviewer 3, the following changes were made (as marked in green).
- There are no information in Abstract on Abeta results. Authors need to add the major results on Ab in the Abstract.
Ans: The major results on Aß have been added in the Abstract section (Lines 27-30 and Lines 32-34).
- It would help readers to grasp the message of figures if authors provide a legend inset representing the corresponding bars instead of numbering in x-axis in Fig. 1, 2 & 4.
Ans: Labeling of X-axis in all Figures have been revised according to the reviewer’s suggestion.
- line 135-136. Check the English grammar.
"These results indicated that the epidermal collagen layer of B12-deficient worms because collagenase-resistant due to the high-level dityrosine crosslinking;"
Ans: The typographical err has been corrected (Lines 133-135).
I hope that you will be able to consider this manuscript for possible publication in International Journal of Molecular Sciences.
Thank you.
Sincerely,
Dr. Tomohiro Bito
Assistant professor
Department of Agricultural, Life and Environmental Sciences,
Faculty of Agriculture,
Tottori University